Evaluating the feasibility of automating dataset retrieval for biodiversity monitoring

http://orcid.org/0000-0002-6416-5731 Fuster-Calvo Alexandre 1 alexfuster7@gmail.com
Valentin Sarah 2
Tamayo William C. 1
Gravel Dominique 1
1 Biology Department, University of Sherbrooke , Sherbrooke, Quebec , Canada
2 Joint Research Unit Land, Remote Sensing and Spatial Information (UMR TETIS), French Agricultural Research Centre for International Development (CIRAD) , Montpellier , France
Provete Diogo
Electronic publication date: 2025 Jan 29
Publication date: 2025
Volume: 13
Electronic Location ID: e18853
Received 2024 Jun 7; Accepted 2024 Dec 20
Copyright: © 2025 Fuster-Calvo et al.
Copyright year: 2025
Copyright holder: Fuster-Calvo et al.
License: This is an open access article distributed under the terms of the Creative Commons Attribution License, which permits unrestricted use, distribution, reproduction and adaptation in any medium and for any purpose provided that it is properly attributed. For attribution, the original author(s), title, publication source (PeerJ) and either DOI or URL of the article must be cited.
License URL: https://creativecommons.org/licenses/by/4.0/

Keywords: Automated data retrieval, Biodiversity monitoring, Data repositories, Ecological data, Machine learning

Funding: NSERC Alliance This study was supported by an NSERC Alliance grant to DG. The funders had no role in study design, data collection and analysis, decision to publish, or preparation of the manuscript.

==============================
Aim

Effective management strategies for conserving biodiversity and mitigating the impacts of global change rely on access to comprehensive and up-to-date biodiversity data. However, manual search, retrieval, evaluation, and integration of this information into databases present a significant challenge to keeping pace with the rapid influx of large amounts of data, hindering its utility in contemporary decision-making processes. Automating these tasks through advanced algorithms holds immense potential to revolutionize biodiversity monitoring.

Innovation

In this study, we investigate the potential for automating the retrieval and evaluation of biodiversity data from Dryad and Zenodo repositories. We have designed an evaluation system based on various criteria, including the type of data provided and its spatio-temporal range, and applied it to manually assess the relevance for biodiversity monitoring of datasets retrieved through an application programming interface (API). We evaluated a supervised classification to identify potentially relevant datasets and investigate the feasibility of automatically ranking the relevance. Additionally, we applied the same appraoch on a scientific literature source, using data from Semantic Scholar for reference. Our evaluation centers on the database utilized by a national biodiversity monitoring system in Quebec, Canada.

Main conclusions

We retrieved 89 (55%) relevant datasets for our database, showing the value of automated dataset search in repositories. Additionally, we find that scientific publication sources offer broader temporal coverage and can serve as conduits guiding researchers toward other valuable data sources. Our automated classification system showed moderate performance in detecting relevant datasets (with an F-score up to 0.68) and signs of overfitting, emphasizing the need for further refinement. A key challenge identified in our manual evaluation is the scarcity and uneven distribution of metadata in the texts, especially pertaining to spatial and temporal extents. Our evaluative framework, based on predefined criteria, can be adopted by automated algorithms for streamlined prioritization, and we make our manually evaluated data publicly available, serving as a benchmark for improving classification techniques.

Introduction

Despite significant progress in establishing global biodiversity databases (e.g., genetic data in GenBank or abundance data in BioTime; Dornelas et al., 2018) and monitoring systems (e.g., the Group on Earth Observations Biodiversity Observation Network (GEO-BON, www.geobon.org) and the Global Biodiversity Information Facility (GBIF, www.gbif.org)), major challenges remain in collecting, cataloging, integrating, and analyzing the vast and diverse datasets needed for comprehensive biodiversity assessments. Historical and contemporary biodiversity data, especially for understudied taxa and less-explored regions, remain difficult to locate and utilize (Jetz, McPherson & Guralnick, 2012; Conde et al., 2019). Even when available, these datasets often fail to capture the full complexity of global biodiversity patterns and dynamics. This shortfall is compounded by the complexity of integrating data across different infrastructures and inefficient data management practices by authors and publishers, creating a significant barrier to advancing biodiversity research and conservation (Chavan & Penev, 2011; Hortal et al., 2015).

A parallel challenge is the ever-increasing volume of data published annually (Hendriks & Duarte, 2008; Stork & Astrin, 2014). As research output accelerates, managing and keeping pace with the expanding wealth of information becomes increasingly challenging. This is largely because the labor-intensive nature of locating and evaluating the pertinence of data within scientific publications for integration into global databases remains a predominantly manual process (Guralnick & Hill, 2009; Wen et al., 2017). The task is further exacerbated by the rapid acceleration in research output, rendering it increasingly impractical to maintain real-time coverage.

In addressing the pressing issue of data scarcity in biodiversity studies, the development of automated systems capable of identifying relevant datasets from diverse sources may well mark a pivotal turning point. These systems bridge text-mining techniques with the interdisciplinary field of natural language processing (NLP) in computer science, integrating methodologies from linguistics, computer science, statistics, and artificial intelligence. Toolsets commonly employ frequency analysis, rule-based algorithms, or artificial intelligence methods (Farrell et al., 2024).

While automatic content analysis is still in its incipient stages for ecological studies (Nunez‐Mir et al., 2016), it has already demonstrated success in streamlining literature reviews (Heberling, Prather & Tonsor, 2019; McCallen et al., 2019), retrieving fossil data (Kopperud, Lidgard & Liow, 2019), monitoring data for endangered species (Kulkarni & Di Minin, 2021), and detecting species co-occurrences and interactions (Farrell et al., 2022). Cornford et al. (2021) showcased the effectiveness of such approaches in identifying relevant articles for specific databases. Their research highlights the capability of algorithms, trained using data from two distinct databases, to analyze a collection of articles. Impressively, these algorithms can discern between relevant and irrelevant articles for these databases with an accuracy rate exceeding 90%, all based solely on the content of titles and abstracts. Recently, prompt-based approaches leveraging large language models such as GPT have demonstrated promising effectiveness in extracting biodiversity data, offering further advancements in automated systems for biodiversity research (Castro et al., 2024).

While these findings mark significant progress, further refinements are essential to realize truly effective automatic retrieval systems. In particular, an automated system should extend beyond merely distinguishing between relevant and non-relevant publications; it should also have the capacity to assign varying degrees of relevance based on metadata, hence aiding in prioritizing the most pertinent studies and expediting their integration into databases. At the same time, the growing trend among scientific journals to mandate the deposition of data used in publications into online repositories reflects a progressive move toward fostering openness in science. Consequently, repositories are anticipated to witness exponential growth, encompassing an ever-expanding volume of published data. In this context, we assess the feasibility of automating the retrieval and classification of datasets from the Dryad and Zenodo repositories. We first introduce a flexible classification framework designed to assign relevance levels to datasets for biodiversity monitoring programs based on features extracted from publication text. This system is adaptable, accommodating both global attributes and specific database requirements. Using this framework, we manually annotated the extracted datasets and then evaluated the performance of a supervised classifier in predicting dataset relevance, using the descriptive textual content of the dataset as input. The automatic approach depends heavily on the availability of explicit metadata in the text, which serves as features for the classification algorithm (such as data type and spatio-temporal range). Our analysis, therefore, examines the distribution of relevant metadata across various sections of dataset publications—such as repository texts and the dataset itself—providing key insights into limits that may hamper the effectiveness of automated search algorithms. Additionally, we compared datasets sourced from Semantic Scholar articles to those from repositories, highlighting the strengths and limitations of each.

Case study-biodiversity monitoring in Quebec

The province of Quebec has recently introduced a dedicated national geographic information system aimed at biodiversity monitoring, known as Biodiversité Québec (https://biodiversite-quebec.ca). Our study is tailored to evaluate the integration of relevant data into this information system, which is designed to play a pivotal role in influencing decision-making processes. Additionally, by situating our research within this dataset, we can showcase the design of specific criteria that are not only globally applicable but also regionally relevant, addressing the unique circumstances at a regional scale. In the context of Quebec, and more broadly in Canada, biodiversity records are particularly affected by spatial bias, correlated to the South-North human population density gradient. Thus, tundra-type ecosystems and northern territories are very poorly documented and data from these regions should have higher priority. This information system leverages the ATLAS data infrastructure, a relational database for the aggregation of biodiversity observations and time series for the province of Quebec. It standardizes various data types (abundances, occurrences, surveys, population time series, and taxonomy) for integration into monitoring and modeling workflows. Data is sourced from open science repositories (e.g., GBIF, eBird, iNaturalist, Living Planet Database) and direct partnerships with local and national organizations. The database is undergoing active expansion, currently aggregating over 53 million occurrences, covering over 23 thousand species from 616 data sources.

Methods

Our methodology involved three key steps. First, we automatically retrieved datasets from repositories using search algorithms that interact with application programming interfaces (API). Second, we manually annotated each dataset, identifying and recording critical features—such as data type, spatiotemporal extent, and taxon information—to evaluate its relevance for biodiversity monitoring. Based on these features, we developed a classification system that assigns the relevance level of datasets for biodiversity monitoring and the ATLAS database. Finally, we applied this classification to evaluate a supervised approach for detecting relevant datasets.

Corpus retrieval

Among the available datasets repositories used to store scientific data for biodiversity research (Nature, 2024), we selected two generalist ones, Dryad and Zenodo. Because Zenodo now hosts a preservation copy of Dryad datasets, it enables the retrieval of datasets from both repositories through a single API. We restricted the number of datasets repositories to two to limit the amount of manual evaluation. The approach is easily transferable to other repositories providing a search API, such as FigShare. To search and extract relevant information, we created a list of queries to interact with Zenodo’s API. The queries, executed in December 2022, were formulated using Boolean “and” operators between keywords selected manually (Fig. 1). The keywords correspond to data type or method corresponding to targeted Essential Biodiversity Variables (EBVs) data (e.g., “presence-only”, “abundance”). We added “Québec/Quebec” to each query to target Quebec-relevant datasets. The API processed each search query, ensuring that all specified keywords appeared within the repository pages, including the title, abstract, or associated metadata. Additionally, the search utilized a Zenodo filter for publications of resource type “dataset”. The detailed approach (queries and code) is provided as Supplemental Material (Fig. S2). Once relevant records were identified, key details such as titles and keywords were extracted and organized in a structured format.

Figure 1 Queries performance retrieving relevant datasets from repositories (Dryad and Zenodo).

(A) Number of publications and counts of relevant categories per query. Colors indicate relevance categories: H (High), M (Moderate), L (Low), and X (Negligible). (B) F scores, precision, and recall metrics for each query. All queries also account for the word “Quebec”.

Dataset manual annotation

Each query generated multiple datasets, all of which were evaluated according to pre-established annotation guidelines. These guidelines encompassed all variables of interest and criteria for assigning feature categories (Table 1; refer to the explanations below and consult the complete annotated dataset in the Supplemental Materials).

Table 1 Manually evaluated features from retrieved datasets.

Feature	Type	Example	
EBV data type	Categorical	Abundance	
Geospatial information	Continuous	Sample coordinates	
Spatial range	Continuous	100.000 km2	
Temporal range	String	From 1999 to 2008	
Temporal duration	Continuous	9 years	
Taxons	String	Black-legged tick	
Referred dataset source	String	Ministère des Ressources naturelles et des Forêts	
Dataset location	Categorical	Repository	
Dataset type location	Categorical	Abstract	
Geospatial information location	Categorical	Article text	
Spatial range location	Categorical	Abstract	
Temporal range location	Categorical	Dataset	
Temporal duration location	Categorical	Article text	

Dataset features

We manually identified dataset features essential for evaluating data type categories, spatiotemporal extent, taxon numbers and identities, and other specific attributes. We also recorded where the features were located because it influences the feasibility of automated retrieval. Locations include the abstract or additional text within the repository page (referred to as repository text), the source publication text (i.e., the article), or within the dataset itself.

Data type categories were assigned following the EBVs data type classification (see EBVs data categories and synonyms in Table S1). Notably, we marked instances when data were time series, as these held unique significance. We quantified the frequency of EBV-related terms in titles and abstracts using the “str_detect” function from the “stringr” R package, ensuring we accurately identified relevant features.

Geospatial data in the datasets were provided at varying levels of precision. We classified this information into distinct categories, ranked from most to least precise: sample, site, or range coordinates, species distribution models (SDMs; thereafter “distribution”), geographic features (e.g., Mont Mégantic), administrative units, maps, and site IDs (where sampling sites were identified, but precise locations were known only to the authors).

Dataset relevance assignment

Datasets were categorized as “High,” “Moderate,” “Low,” or “Non-relevant” based on their relevance to biodiversity monitoring. Our classification system relied on two groups of criteria. The first group, termed Main Classifiers, captured fundamental features of biodiversity data, including data type, temporal extent, and spatial extent. This categorization was informed by the literature on Essential Biodiversity (Pereira et al., 2013; Schmeller et al., 2018; Jetz et al., 2019) and the importance of temporal and spatial dimensions. For example, temporal extent was categorized as follows: less than 3 years as Low, 3 to 10 years as Moderate, and more than 10 years as High (see Table 2). Each dataset was assigned a relevance category for each Main Classifier. For instance, a dataset could be classified as Low for data type (e.g., presence-only), Moderate for temporal extent (e.g., 5 years), and Moderate for spatial extent (e.g., 10,000 km2). The overall relevance score for a dataset was computed using a majority rule. In this example, the dataset’s relevance by Main Classifiers would be Moderate (Low, Moderate, Moderate = Moderate). Datasets deemed “Non-relevant” provided no valuable information for biodiversity monitoring. In case of non-majority value, we create decision rules as indicated in Table S2. By default, the relevance score of the dataset type was selected as the final relevance category, penalized if the spatio-temporal relevance was Non-relevant, Low, or Moderate.

Table 2 Evaluation criteria used to assign the relevance category to the datasets.

Classifier group	Classifier	Relevance assignment	Example	
Main classifiers	Data type	High: Abundance, density, EBV genetic analysis.	L	
Moderate: Distribution, presence-absence.	
Low: Presence-only, relative abundance, species richness, non-EBV genetic analysis.	
Spatial range	High: >15.000 km2.	M	
Moderate: 5.000 to 15.000 km2.	
Low: <5.000 km2.	
Temporal range	High: >10 years.	M	
Moderate: 2 to 10 years.	
Low: <1 to 2 years.	
Assigned relevance by main classifiers (criteria: majority vote)	M	
Modulators	Multispecies	>10 species.	FALSE	
Threatened species	Assigned a threatened level by IUCN.	TRUE	
New species to science	A first description of a species.	FALSE	
New regional species	A first record of a species in the region.	FALSE	
Priority area*	Data from a geographic area of special interest.	FALSE	
Assigned relevance by Main Classifiers + Modulators (criteria: if any TRUE, relevance by Main Classifiers increases one category)	H	
Note:

* For our study case, the ATLAS database, we prioritize areas from Northern Quebec.

First, the main classifiers determine whether the dataset relevance is “High”, “Moderate”, “Low”, or “Non-relevant” based on a majority vote (see Table S2 in case of disagreement), and then Modulators increase one relevance category if any of them is present (e.g., “Low” to “Moderate” but not “Low” to “High”). Modulators do not affect datasets classified as “Non-relevant” by the Main Classifiers.

The second group of criteria consisted of Modulators—features that, while less critical than Main Classifiers, added context that could slightly modify a dataset’s relevance. Modulators addressed regional or specific database needs, such as a first record for a given taxon in a region or data from under-sampled areas. For example, in the context of Quebec’s biodiversity monitoring strategies, data from northern regions (the territory that extends north of the 49th parallel and north of the St. Lawrence River and the Gulf of St. Lawrence) are prioritized due to the lack of studies in these areas, making the north-south sampling bias a key modulator for Quebec-focused datasets. Modulators then allow flexibility to accommodate specific databases’ requirements.

We introduced the following Modulators, each of which could be marked as true or false: multispecies (containing data for more than 10 species), presence of threatened species (as assigned by the IUCN), new species to science, new species to Quebec (explicitly mentioned in the title or abstract), and location in northern Quebec (Table 2). For multispecies datasets, only the north-south modulator was noted. If a dataset met any of these modulator criteria, its relevance category based on Main Classifiers would increase by one level. For instance, if the example dataset mentioned earlier included data from an endangered species, the “Threatened species” modulator would apply, and its relevance would increase from Moderate to High (Table 2). Only relevant datasets (those classified as Low, Moderate, or High by the Main Classifiers) could be upgraded by Modulators.

Extension to scientific articles

We expanded our dataset retrieval beyond repositories to include articles sourced from Semantic Scholar, covering the period from 1980 to 2022. Semantic Scholar relies on a search relevance algorithm. Briefly, the provided query is to Elasticsearch, where about 190 million scientific articles are indexed. We provide in Table S3 the approximate number of results matching with the studied queries obtained through Semantic Scholar websites, ranging from 23,200 to 3,490,000 results. Through the API, the top 1,000 results are re-ranked by a machine learning ranker. To ensure a manageable analysis, we therefore had to limit the retrieved publications to a maximum of 50 positives (top 50 found publications). Due to potential API request limitations, some queries retrieved more publications than others, with an average of 23 publications per query. Subsequently, we assessed the relevance of the datasets and publication years in the same manner as with repositories, facilitating comparisons between these two sources.

Automatic relevance classification

To evaluate the feasibility of automating the relevance classification process, we trained binary classification algorithms to predict the relevance. As input, we use the text from the title and repository descriptive content of the dataset (for Zenodo and Dryad datasets) and from the title and abstract (for Semantic Scholar datasets). The text was first cleaned by removing special characters and converted to lowercase. We then applied different text pre-processing steps, including stop-words removal, lemmatisation and selection of n-grams range, as detailed in Supplemental Material S1. Stop-words are frequently used words which generally do not hold any significant semantic value, such as ‘the’, ‘for’, etc. We filtered them using a predefined list of 39 English stop words from the python NLTK library (Bird & et Edward, 2004). N-grams are contiguous sequences of n words or characters from a text. Unigrams are sequences of a single word (e.g., ‘ecology’), and bigrams are sequences of two consecutive words (e.g., ‘arctic ecology’). Taking into account bigrams might help capture local relationships between words. We represented the texts either with unigrams alone or with unigrams and bigrams, using the ngram_range parameter of the TfidfVectorizer() function from python scikit-learn library (Pedregosa et al., 2011). Lemmatisation converts each word into a base form (i.e., lemma), to ensure that all variations (e.g., singular and plural form) of a word are represented uniformly. We used the WordNetLemmatizer from the NLTK library.

To transform the text into numerical form, we employed a frequency-based representation, combining bag-of-words with TF-IDF (term frequency-inverse document frequency) weighting. In the bag-of-words representation, each document is represented by an n-dimensional vector where each component represents the absence or presence of a word i in the document d and n is the vocabulary length (Zhang, Jin & Zhou, 2010). TF-IDF adjusts this by assigning higher weight to terms that are more frequent in a document relative to their frequency across the collection of documents (Salton & Buckley, 1988). We used the TF-IDF vectorizer function from the scikit-learn library (Pedregosa et al., 2011) and ignored terms with a document frequency lower than 3.

We compared three classifiers: Logistic Regression (Ifrim, Bakir & Weikum, 2008), Random Forest and a linear Support Vector Machine classifier (SVM) (Joachims, 1998). For all classifiers, we used the implementation from the scikit-learn library. We set the class_weight parameter to « balanced », which adjusts the class weights inversely proportional to their frequencies (i.e., classification performances of the majority class are penalized during the training), addressing the class imbalance in our data.

Due to the small size of the corpus, we merged the non-relevant and low-relevance classes (treated as non-relevant, the negative class), and the medium-relevance and high-relevance classes (treated as relevant, the positive class).

Evaluation

Queries

We analyzed the performance of each query by noting the amount of retrieved, not found, and not accessible datasets and their relevance categories. For each dataset, we the precision and recall metrics and their harmonic mean, known as F-score (Goutte & Gaussier, 2005), as follows:

Fscore=2⋅Precision⋅RecallPrecision+Recall

where:

Precision=TruePositiveTruePositive+FalsePositiveRecall=TruePositiveTruePositive+Falsenegative

Positive refers to those datasets assigned a high or moderate relevance, and negative to low or negligible relevance.

Features and accessibility

Our assessment encompassed both the content and accessibility of datasets. To gauge the spatiotemporal extent and the representation of Essential Biodiversity Variable (EBV) categories within the datasets, we conducted visual inspections to discern the alignment between dataset duration and spatial range frequencies, and their assigned relevance categories.

In parallel, we assessed the accessibility of features, a crucial factor for automated retrieval. To this end, we tallied the occurrences of feature locations within the datasets, distinguishing between repository text, articles, and dataset contents, for dataset type, temporal, spatial, and taxon features. Additionally, we cataloged the frequency of occurrences for each geospatial information category to gain insights into the level of precision provided in this regard.

Semantic scholar-repositories evaluation

We evaluated the textual content of abstracts retrieved from Semantic Scholar and the repositories, examining various aspects This entailed evaluating the relevance, number, and accessibility of datasets obtained from both sources. To delve deeper into the temporal dimension, we quantified the frequencies of publication years for the datasets.

Automatic relevance classification

We used five-folds cross-validation and calculated the average precision, recall and F-score as previously defined. We then evaluated the classifier’s performance in predicting the relevance of the Main Classifiers, as well as the relevance of the Main Classifiers combined with the Modulators.

Results

Datasets annotation evaluation

Out of the initial 161 datasets retrieved through our queries, 57 were subsequently excluded for various reasons: 37 due to incorrect locations, five categorized as laboratory studies, and 13 for miscellaneous reasons. Notably, many datasets with incorrect locations resulted from the matching of the keyword “Quebec” with the affiliations of the authors.

From the remaining 104 datasets we kept, the classification based on relevance yielded 89 relevant datasets categorized as either highly, moderately, or low in relevance, which we will refer to as “relevant datasets”: 36 datasets (22%) categorized as highly relevant, 26 (16%) as moderately relevant, 27 (17%) as having low relevance, and 15 (9%) as non-relevant.

Queries

The most simple query, “species”, is the one showing the highest performance (F-score = 0.28), followed by “population + species” (F-score = 0.23) and “sites + species” (F-score = 0.16). “Occurrence + species” was the query with the highest precision (0.39) (Fig. 1). The mean overlap between queries was 11% (Fig. S1).

Features

Among the 89 relevant datasets, presence-only data emerged as the most common EBV data category, encompassing 31 publications (29%), followed by genetic and abundance data with 27 (25%) each. In contrast, data from species distribution models (SDMs), referred to as “distribution” data, was the least common, featured in only four publications (3%) (Fig. 2C). The temporal ranges of these datasets span from the 1930s to the present, although the majority fall within the last two decades (Fig. 2A). Notable outliers include Favret et al. (2020), which offers data on Odonata specimens from various entomological collections dating back to 1875. On average, the temporal duration was 10.9 years, ranging from less than 1 year to 50 years (Fig. 2B). Short-term studies, less than 1 year in duration, constituted the most common category, accounting for 19% of the datasets. A total of 12 datasets (13%) contained time series data (Fig. 2A). Spatial extents varied widely, ranging from 0.2 to 24.706.834 km2, with a mean of 1,369,232 km2: 25 datasets (28%) covered less than 5,000 km2, 9 (10%) fell between 5,000 and 15,000 km2, and 27 (30%) exceeded 15,000 km2 (Figs. 2B and 2D).

Figure 2 Features of retrieved datasets from repositories Dryad and Zenodo and results of the relevance evaluation.

(A) Temporal duration and ranges. Red range bars indicate datasets with time series data. The first dataset extends to 1875. (B) Duration, spatial rage, and relevance category. (C) Publication counts by EBV data types. (D) Spatial range counts in the low, medium, and high range categories, respectively. (E) Publication counts by taxa and relevance categories: the left panel shows species-level studies, which contain data for 1 to 10 species, whereas the right panel shows community level studies with data for more than 10 species. Letters H, M, and L correspond to “High”, “Moderate”, and “Low” dataset relevance categories, respectively.

The datasets cover a diverse array of taxa, spanning 13 classes (not counting those within zooplankton). Among these, 66 (74%) datasets were associated with one to ten species, with mammals (21), fish (13), birds (11), and angiosperms (10) being the most frequently represented. In contrast, 23 (26%) datasets pertained to communities comprising 10 to 439 species, with an average of 58 species per dataset. Notably, datasets of this nature were more prevalent for plants (13 datasets) and insects (six datasets) (Fig. 2E).

Features accessibility

Within the 89 relevant datasets, at least one comprehensive metadata (features) belonging to Main Classifiers was automatically accessible for 81 of them (91%), typically within the repository page’s abstract or additional text. These encompassed explicit mentions of temporal range in 26 publications (29%), temporal duration in 14 publications (16%), and spatial range in seven publications (8%) (Fig. 3A). For species-level studies (involving 1 to 10 species), species names were consistently present in the title or abstract. Additionally, dataset types or synonyms were included in the title for 22 of them (25%) and in the abstract for 80 (90%).

Figure 3 Location of features (i.e., metadata) in the publication spaces.

(A) Spatiotemporal features and (B) geospatial features.

Only two publications contained all these features together in the repository text. Furthermore, a total of five (6%) did not explicitly report the temporal range, 30 (34%) the spatial range, and 60 (67%) the data duration in any location (repository text, source article, appendix, or dataset) (Fig. 3A).

Geospatial information, essential for biodiversity monitoring, was unavailable within the repository text in 65 publications (73%), and no location data, including the dataset, was provided for 33 (37%) of them (Fig. 3B). Site IDs were given in the dataset of 23 (26%) publications, but in most cases, it required consulting a map in the article, with no specific coordinates, to interpret them.

Semantic scholar

Our queries yielded 254 datasets from Semantic Scholar without overlap, of which 62 were excluded due to incorrect locations (33), unrelated fields outside of biodiversity (21), experimental data (six), or microbial communities (two). Among the valid datasets, classification by relevance resulted in 36 (19%) high, 26 (14%) moderate, 27 low (14%), and 15 (8%) non-relevant datasets, alongside 28 (15%) inaccessible datasets and 84 (44%) publications without datasets (Fig. 4A). A comparison of publication years between Dryad and Zenodo vs. Semantic Scholar revealed that retrieving from repositories yielded datasets published from 2010 onwards while retrieving from Semantic Scholar included older datasets dating back to 1981 (Fig. 4B).

Figure 4 Comparison between semantic scholar and repositories (Dryad and Zenodo).

(A) Relevance and accessibility of datasets. (B) Temporal depth of retrieved publications.

Furthermore, while one highly relevant dataset, the Neotoma Paleoecology Database (www.neotomadb.org), was referenced in publications extracted from repositories, we identified a total of six highly relevant datasets referenced in articles retrieved from Semantic Scholar. These datasets originated from sources such as the Canadian National Forest Inventory, the Canadian Wildlife Service, and the Québec Ministry of Environment and Wildlife.

Automatic relevance classification

The classification of the Main Classifier shows moderate overall performance, with a weighted F-score ranging from 0.63 to 0.68 depending on the pre-processing steps and classifiers (Supplemental Material S1). Pre-processing steps had limited impact on the evaluation metrics. We highlight in Table 3 the detailed performances of the Logistic Regression algorithm which obtained the highest F-score (0.68) when applied after stop-words removal and considering only unigrams. We note that the classifier struggles to correctly identify relevant datasets, as indicated by the low recall of 0.44—meaning it correctly detects less than half of the relevant datasets. The Main Classifier + Modulators relevance prediction overall performance in terms of F-score ranged from 0.53 to 0.61. The performances metrics shown in Table 3 correspond to the use of Random Forest classifier on the lemmatised text, using both unigrams and bigrams. The recall of the relevant texts reaches 0.71.

Table 3 Performances metrics of the classification to predict the Main Classifier (left) and the Main Classifier + Modulars (right) classes (relevant/non-relevant), in terms of precision, recall and F-score.

	Main classifier relevance	Main classifier + Modulators relevance	
	Classifier: logistic regression, pre-processing: stop-words removal, unigrams	Classifier: random forest, pre-processing: lemmatised text, no stop-words removal, unigrams and bigrams	
	Precision	Recall	F-score	Precision	Recall	F-score	
Relevant	0.57	0.44	0.50	0.62	0.71	0.67	
Not relevant	0.73	0.82	0.77	0.59	0.49	0.53	
Weighted average	0.68	0.69	0.68	0.61	0.61	0.61	

We outline the most important features identified by the Support Vector Machine and Random Forest models in Tables S4 and S5. The features specific to certain taxa (e.g., tree, fish, carex) or ecosystem (e.g., arctic, boreal forest, peatlands) show higher weights, whereas vocabulary describing dataset types in a more precise way (e.g., range, distribution, specimen, etc.) and responses to climate change (e.g., climate, change) had lower weights. We also noted spurrious correlations in Random Forest results with unrelevant textual features such as adverbs (e.g., particularly) or figures (e.g., 16).

Discussion

We have introduced a classification system, designed to serve as a logical framework for automated algorithms, streamlining the evaluation process by categorizing data based on its relevance. This system draws upon globally applicable biodiversity classifiers, making it adaptable to various data types, while also allowing for the incorporation of dataset-specific nuances. To support future advancements, we have also published our high-quality, manually annotated dataset (available alongside this article), providing a benchmark for developing new classifiers. Using this dataset, we trained an automated algorithm that showed moderate performance in identifying relevant datasets. Despite this, our study highlights both the feasibility and the potential value of automating the retrieval and classification of biodiversity datasets, offering a strong foundation for further refinement and improvement in this area.

Assigning relevance categories, whether through our system or alternative approaches, necessitates a meticulous analysis of features (i.e., metadata) within the publication text. Our study highlights that this task poses a considerable challenge for automated processes, often stemming from the absence or scarcity of these features and their sparsity across different sections of the publication (repository page, dataset, article, Supplemental Materials). This challenge is particularly pronounced in the case of spatio-temporal features, which are pivotal for guiding relevance assessments. Furthermore, our study demonstrates that repositories present a valuable source of readily accessible, publicly available data, surpassing scientific articles in terms of speed and efficiency for data retrieval. However, we also emphasize the significance of designing automated processes for data extraction from articles. These offer a substantial temporal depth of publications and serve as gateways that can guide researchers to other pertinent and valuable data sources.

Remarkably, a high percentage of datasets contained genetic data, an area where greater collection efforts have been advocated (Hoban et al., 2021, 2022). Moreover, following presence-only data, abundance data was the most frequently encountered, being information that can aid in constructing time series and elucidating population trends. These are promising findings for the repositories’ potential as pivotal resources for automating the retrieval of biodiversity data.

We have devised a comprehensive classification system aimed at assigning relevance categories to datasets, grounded in a set of criteria. We advocate for the adoption of similar schemes by automated dataset detection algorithms to aid in prioritizing datasets and maintaining pace with the relentless surge in published data. A critical aspect of our system involves the distinction between criteria as either Main Classifiers or Modulators: the former encompasses fundamental, overarching aspects pertinent to biodiversity data on a global scale, while the latter encompasses secondary criteria that retain significance, potentially at regional scales. This distinction enabled our framework to underscore the vital importance of data concerning plants and animals in boreal (e.g., Lait & Burg, 2013; Thiffault et al., 2015; Martin et al., 2022) and arctic ecosystems (e.g., Leblond, St-Laurent & Côté, 2017; Lamarre et al., 2018; Chagnon, Simard & Boudreau, 2021), which face heightened vulnerability and remain underexplored in Quebec, along with the inclusion of new species (Anderson & Lendemer, 2016). Our assessment, however, has revealed a significant challenge in identifying the information essential for automated algorithms to assign categories of relevance to publications. This challenge stems from the absence and dispersion of critical features throughout various sections of the publication, making it particularly problematic for detecting Main Classifiers related to temporal and spatial extents. In the majority of datasets we examined, these details were either entirely missing or exclusively located within the article text, neglecting inclusion in the abstract or repository page.

The automated classification process, while showing moderate performance, provided valuable insights into the key features driving the relevance predictions. We had anticipated higher weights for more precise vocabulary related to dataset types, and the fact that features specific to certain taxa and ecosystems were identified as more important suggests some degree of overfitting. The limited size and geographic focus of the training dataset likely introduced biases toward particular taxa or ecosystems, potentially leading to spurious correlations and explaining the model’s lack of generalization to the validation dataset. However, the identification of key terms such as “changes,” “genetic,” and “population” indicates that relevant descriptors were detected by the classifier. This suggests that expanding the dataset or applying data augmentation methods (Tang, Kamei & Morimoto, 2023) could improve the performance of the supervised approach. An avenue for improvement would be to use a list of taxa and ecosystems-related terms and filter them from the vocabulary used by the model to focus on the more meaningful terms. In the context of data scarcity, an alternative approach would be to directly extract key features from the texts (e.g., data type, temporal extent, etc.) without relying on a supervised approach. These features may be combined a posteriori in the same framework as described for manual evaluation. Eventually, we could adopt the strategy of Cornford et al. (2021) who automatically created a set of irrelevant documents by extracting 125,000 articles about ecology from a national database. This strategy allows for a drastic increase in the training dataset, but as the set of irrelevant documents is not manually evaluated, there is a risk that they contain irrelevant elements. The different pre-processing steps and classifiers had a limited impact on the performance metrics. We, therefore, conclude that either our dataset is too small to enable a robust comparison, or the lack of availability of important features in the textual content is an incompressible limit to the model’s performance.

We only evaluated classification performance with a traditional bag-of-words representation and machine learning algorithms, specifically logistic regression, Random Forest and a linear support vector machine (SVM). Recently, pre-trained large language models (LLMs), such as bidirectional encoder representations from transformers (BERT) (Devlin, 2019), which require minimal feature engineering, have gained widespread use in many natural language processing tasks. While they do not always outperform bag-of-words approaches (Wahba, Madhavji & Steinbacher, 2022), including them in a more comprehensive assessment would be valuable. LLMs’ ability to better account for semantics may improve the detection of dataset types through implicit descriptors. However, like the bag-of-words approach, they would still be limited by the absence of spatiotemporal information. Therefore, search algorithms should be designed to scan all parts of a publication, including the dataset itself, to capture these essential features.

This underscores the urgency of applying general and standardized frameworks within publication guidelines to ensure the necessary metadata is provided for automatic detection and extraction. Presently, various global frameworks and initiatives, such as the FAIR Data Principles and GBIF standards, advocate for accompanying data with comprehensive metadata using standardized formats such as Darwin Core (Wieczorek et al., 2012). However, the way metadata is currently presented in scientific articles and repositories often falls short of meeting these needs, posing a significant obstacle to retrieving relevant biodiversity data (Jones et al., 2019; Löffler et al., 2021).

Our findings highlight that repositories like Dryad and Zenodo, as well as scientific literature search engines such as Semantic Scholar, offer distinct advantages and disadvantages when it comes to automated data retrieval. Sources of scientific literature may exhibit a relatively high percentage of publications that lack data or render it inaccessible as it was observed for Semantic Scholar, which can potentially pose challenges when retrieving datasets and assessing their relevance. Filtering whether or not an article is associated with a dataset would allow to filter out irrelevant articles before the relevance assessment. In practice, we believe that this task would not be as trivial, as datasets are sometimes provided through a link towards a repository, or supplemental material. Such information is not accessible through the title and abstract only and would require retrieving the entire article content. Moreover, the quality of the retrieval also depends on Semantic Scholar’s relevance ranking model. Since we restricted our evaluation to 50 results per query, our evaluation does not reflect the intrinsic value of the Semantic Scholar database. Increasing this limit would contribute to a more robust evaluation. Conversely, repositories offer readily accessible datasets and prove to be highly efficient sources for data retrieval. However, repositories may exhibit a limited temporal depth of datasets, typically spanning only the past decade, given their contemporary nature. In contrast, sources of scientific literature have the potential to provide a more extensive historical dataset archive. Moreover, articles may refer to and cite other data sources, such as the six distinct highly relevant databases we found referenced in retrieved articles, thereby facilitating the identification of unknown relevant datasets. These substantial differences underscore the importance of conducting data retrieval from both types of sources to ensure a comprehensive approach.

In the realm of automated algorithm development, manual evaluations, such as the one conducted in this study (publicly available; see Data Availability statement), will remain invaluable. They serve both as training data for automated algorithms to learn to identify relevant datasets and as a crucial benchmark that enables the assessment of automated processes by drawing comparisons with the results from manual evaluations. Furthermore, it’s imperative to acknowledge the need for specialized strategies when dealing with diverse data sources. For instance, one we did not assess here is the increasingly digitized collections of museums, which might require specific automated search and evaluation approaches. An exciting avenue for future research lies in these sources, which harbor invaluable historical data often absent from contemporary global information systems or databases (Graham et al., 2004; Guralnick, Hill & Lane, 2007; Page et al., 2015; Wen et al., 2015).

While the development of literature classification algorithms is advancing vertiginously, especially with artificial intelligence (AI) systems (see Google Gemini, www.deepmind.google), it is imperative to recognize that the challenges surrounding information structure and metadata organization within scientific literature persist regardless of technological evolution. The effectiveness of automated systems relies not only on the sophistication of algorithms but also on the clarity and consistency of metadata standards, the accessibility of data repositories, and the interoperability of databases. Our work not only identifies significant challenges for forthcoming automated algorithms tasked with dataset retrieval for biodiversity but also underscores the relatively surmountable nature of these challenges. These foundational issues transcend the current state of AI technology and are central to ensuring the long-term viability and utility of automated systems for biodiversity data retrieval. As such, efforts to address these structural challenges as well as standardized schemes for prioritization such as the one we proposed, must remain a priority alongside advancements in AI capabilities. To move forward effectively, prioritizing the applications of global frameworks and guidelines that streamline the workflow for automated data retrieval systems is essential. These initiatives carry the transformative potential to reshape our data acquisition capabilities profoundly, greatly bolstering our capacity for biodiversity monitoring and informed decision-making, with far-reaching positive ecological implications.

Supplemental Information

Supplemental Information 1 EBV data categories used in our framework and its synonyms.

Supplemental Information 2 Main classifier relevance assignment in case of disagreement.

H: highly relevant, M: Moderate L: low, X: non-relevant

Supplemental Information 3 Number of results returned through Semantic Scholar website for each query (temporal range: 1980 - 2022).

Supplemental Information 4 Most important features (top 15) of automatic classifiers for Main Classifier relevance by Support Vector Machine (stop-word removal and unigrams/bigrams selection).

Supplemental Information 5 Most important features (top 15) of automatic classifiers for Main Classifier relevance by Random Forest (stop-word removal and unigrams/bigrams selection).

Supplemental Information 6 Approach to provide queries into APIs.

To ensure that keyword boolean combinations are properly taken into account in Zenodo’s and Semantic Scholar’s API, we first evaluated their behaviour in the management of three query formats: the accents (e.g. “Québec” or “Quebec”), the plural forms (“inventory” or “inventories”) and the order or the keywords in the boolean query. Zenodo’s API is sensitive to both plural forms and accents (i.e. the query “inventory+Quebec” does not retrieve the same results as “inventories+Quebec” nor as “inventory+Québec”). We interacted with the API using the request library, including the creation of plural and accent variations for Zenodo, and request formatting.

Supplemental Information 7 Percentage overlap between search queries used on Zenodo API.

Supplemental Information 8 Supplementary material about relevance classification.

We thank Vincent Beauregard for his feedback on the algorithms and the ATLAS database. We sincerely thank the reviewers for their insightful comments and suggestions, which have greatly improved the quality and clarity of this manuscript. We acknowledge the use of ChatGPT to help improve the fluency and grammar of the manuscript text.

Additional Information and Declarations

Competing Interests

The authors declare that they have no competing interests.

Author Contributions

Alexandre Fuster-Calvo conceived and designed the experiments, performed the experiments, analyzed the data, prepared figures and/or tables, authored or reviewed drafts of the article, and approved the final draft.

Sarah Valentin conceived and designed the experiments, performed the experiments, prepared figures and/or tables, and approved the final draft.

William C. Tamayo performed the experiments, authored or reviewed drafts of the article, and approved the final draft.

Dominique Gravel conceived and designed the experiments, authored or reviewed drafts of the article, and approved the final draft.

Data Availability

The following information was supplied regarding data availability:

The data and script are available at GitHub and Zenodo:

- https://github.com/Alex-Fuster/automated_datset_retrieval.git.

- Fuster-Calvo, A., Valentin, S., C Tamayo, W., & Gravel, D. (2024). Evaluating the feasibility of automating dataset retrieval for biodiversity monitoring. https://doi.org/10.5281/zenodo.13881796.

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
