# Peer review of "Evaluating the feasibility of automating dataset retrieval for biodiversity monitoring"

_PeerJ, doi:10.7717/peerj.18853_

## Round 0.1 · original submission · Major Revisions

Dear authors,

We have now two reviews back on your interesting manuscript. While they acknowledge the high potential of the automatic retrival system, but at the same time both of them point our that the Methods section lacks clarity. For example, both highlight that the "automatic" part of the pipeline is somewhat not clear, since everything seems to have been made by hand. The methods section is indeed extremely confusing and hard to distinguish key aspects of what you did. This section also has to be fully re-written to be more streamlined, straightforward, and short.

As for myself, I believe the introduction needs to be fully re-written, following criticism of R2 and mine below. It simply doesn't present well the very objective of the paper.

I also have some more comments:

1) The introduction is rather long, you can shorten especially the begining (e.g., mentions to EBV and monitoring in general, text mining etc), since these topics are not addressed in your paper. The part on Québec can also be reduced to one paragraph maximum.

L. 133: relevance for what?
L. 207, 214, 215: again, relevance for what?

·

Basic reporting

no comments

Experimental design

The main objective of the manuscript is to carry out a manual evaluation of biodiversity data retrieved from the Dryad and Zenodo repositories in order to access the potential of automatic classifications in various "relevance tiers". The classification system is flexible, allowing researchers to adjust it to their context, and can serve as a foundational baseline for the next automatic classification algorithms in the literature. The authors used the province of Quebec (Canada) as a case study. The authors searched for data to add to the local "Biodiversité Québec" system. In this system, the data is spatially skewed, with the Northern territories and the Tundra ecosystem being the least documented and requiring the highest priority.

The article is well written and of international interest.
My main concern is with the method, which is difficult to understand. I suggest that the authors revise this part to make it clearer how they did the whole process of database recovery and automation.

Validity of the findings

no comments

Additional comments

Methods
The authors need to better detail how they retrieved the data from the repositories.
Lines 190-193: The author mentions that he has built functions that interact with the API, but does not detail these functions and does not give examples
Lines 223-232: The authors do not describe the "modulators" variables, they need to be more specific as they were with the main variables

Other comments attached.

Reviewer 2 ·

Basic reporting

The text is well written, but suffers from a lack of clarity in the Methods (see Additional comments)

Experimental design

It appears that the authors do not do what they claim in the title and introduction of the paper. Instead of evaluating the feasibility of an automated system, the authors present a manually labelled dataset which may be used for such an endeavour in the future.

In some minor instances, the mkethods were not decribed with sufficient details (e.g. the creation of their relevance label).

See Additional comments

Validity of the findings

See Additional comments

Additional comments

This article claims to investigate the potential to develop a ranked classification system to automate the identification and curation of published biodiversity datasets. While the concept and effort are commendable, I cannot easily see where the automated component of the pipeline comes in. This may be due to a lack of detail in the methods, but it appears that the authors generated a set of classifications based on manual labelling of datasets and articles, but they do not describe a model or automated approach to generate these labels. The dataset generated by the authors (manual labels with corresponding features from the article text and metadata) are amenable to fitting and testing the accuracy of a formal automated approach and classification model (e.g. regression / GBM, etc), but the main results of the paper seem to be the results of their search rather than the results of any such model.

I suggest that the authors generate this automated framework (e.g. automate labelling, fit alternative classification models, and validate their approach), as this is what I expected given the title and introduction. Instead this article appears to be creating and presenting a labelled dataset which may be used for future endeavours. In this case, I do not feel the authors have sufficiently evaluated the feasibility of automating dataset retrieval for biodiversity monitoring.

Minor:

Line 187-188: You explain the criteria used for identifying data repositories, but do not describe your search strategy to identify the ones you chose. These may be the only two with an API for the biodiversity domain, but as you have not presented the alternatives which did not fit the criteria, it feels like a post-hoc justification and Zenodo was chosen for purposes of ease.

There is quite a lot of text devoted to broader context (e.g. lines 46-85; 154-163). These could be greatly streamlined to more quickly address the core issues and context of your study.

Lines 261-264: Limiting search results to the top 50 articles per query is at odds with your justification to use automated approaches for literature search and synthesis. I strongly suggest removing this removing this restriction. However, if you do decide to restrict searches, you should provide a description of article databases included in Semantic Scholar, report the number of total results per query and the fraction you were able to retrieve, and describe how their search algorithm ranks searches (e.g. is this based on frequency of terms in abstract and title, or some composite metric which also includes the cited literature – this is the case for the Web of Science “Topic” field, which from personal experience can article ranking to show non-relevant articles which cite relevant content).

Line 266-267: Unclear how your approach “generates a random sampling among queries” if you are using the top search results and have some searches restricted by API request limits.

Line 326-327: There is no mention of searching for Quebec-related data via author affiliations in your methods section. Please detail how you filtered to datasets with data relevant to Quebec, as author affiliation is clearly not always useful, and large spatiotemporal datasets may include data from Quebec but not mention “Quebec” directly in the article (e.g. a survey across Canada or North America may only have Quebec mentioned deep in the raw data, or identified only when plotting geospatial coordinates).

Lines 336-339: Please provide justification for using “species” in all of your search queries. Couldn’t this possibly exclude many relevant studies?

Lines 345-346: I am surprised SDMs only represented 3 publications in your dataset. A quick Google scholar search ("species distribution model" AND quebec) resulted in 690 articles. This points to a potential issue that your search strategy not being general enough to encompass all relevant datasets.

Lines 389: Please outline what the “miscellaneous reasons” included.

Table 2: Unclear how you combine each of the classifiers to get your final “Relevance”. From the table it looks to be the equivalent to the highest relevance category for any of the classifiers. Is this correct? You should provide additional details on the exact calculation as I could not easily find this in the methods text.

---

## Round 0.2 · Major Revisions

I have now received back the comments from the same two reviewers from the last round. While R1 is happy with the way the manuscript looks now and only suggests a few minor adjustment to the text, R2 is a bit more critical, asking for more details in the Methods and suggesting the need to re-run analysis using a different approach.

While I believe R2's comments are relevant and could eventually improve the paper, I will let the authors decide how they want to deal with them.

·

Basic reporting

no comments

Experimental design

no comments

Validity of the findings

no comments

Additional comments

Dear Editor and authors,

The authors have addressed all suggested revisions, resulting in significant improvements to the manuscript. I recommend accepting it for publication, with only a few minor suggestions:

Lines 215-216: Is there a link or citation for the Biodiversité Québec system?
Line 225: What does ATLAS stand for? If it is an acronym, please spell it out at first mention and provide a citation.
Line 237: Is API an acronym? If so, please specify and explain what "API functions" refers to.
Line 505: Fscore
Thank you for the opportunity to review this enhanced manuscript.

Best regards,

Reviewer 2 ·

Basic reporting

See 4.

Experimental design

See 4.

Validity of the findings

See 4.

Additional comments

I reviewed a previous version of this manuscript and am very happy to see that the authors have responded to the majority of my suggestions and concerns. The authors have included information on their classification method (SVM), which unfortunately only performs moderately well at classifying relevant versus non-relevant databases. Based on the description of methods, I think this is due to a few major issues – one is a small training dataset (which can be improved if the authors increase the number of Semantic Search results they include), and potential issues with bias in terms of the taxa and ecosystems represented in their training data. Further, the authors are predicting labels based on a composite measure of relevance in which datasets may be considered relevant based on different criteria. The authors may find improved model performance if directly predicting the components of their composite rather than the overall relevance. I have detailed these concerns below, and suggest some alternative approaches which may improve their model performance. In addition to these, I find that the methods lack detail in some places (e.g. text pre-processing steps of stop word removal and frequency-based filtering). Aside from methodological issues, I still find that the introduction has some unnecessary context, and the explicit link to Quebec biodiversity as a case study is not clear.

Overall, I think this could be a good contribution to the growing literature on text-based classification of ecological research, but the overall approach and methods have much room for improvement.


Major Issues

- Lines 247-250: The arbitrary limitation of search results to 50 per query (which seem to be reduced by half due to API limitations) is a critical limitation of the study. It is unclear how Semantic Scholar’s “machine learning ranker” identifies these top 1000, and how different the top 50 versus ones below this are in terms of relevance. Choosing to restrict to the top 50 is likely leading to omission of relevant datasets, and contributing to a bias in which types of papers are included in your study. I strongly suggest you identify a way to quantify relevance if you choose to do some restriction at this stage. If Semantic Scholar reports a relevance metric per article, this would be an excellent start. If they do not, then you may be able to identify uncertainty in relevance by comparing the relative ranks of articles returned across your different search terms. In lines 335-338 you say that there was fairly little overlap between queries. This is surprising as “species” was included in all search terms. If this represents a rigorous search, this first “species” only search should include all results from other searches. Again, this low overlap indicates that this initial filtering is too restrictive. I understand that restricting to under 50 publications per query makes for a more manageable analysis, but this is greatly restricting the amount of training data you have to build your SVMs. In line 390-391 you say that 44% of articles do not have associated datasets. This points to the ability to more rapidly screen studies by first checking that they do in fact have associated datasets. If doing this filtering first, you could quickly and fairly easily filter out ~440 of the top 1000 articles returned by Semantic Scholar... I touch on this in a later comment, but you have a situation with many more predictors than observations, and this simple act of filtering here may be a major reason why your model shows much lower performance compared to previous work (e.g. Cornford et al. 2022).


- Lines 260-261: Please provide additional details on how you identified common stop words. In many NLP analyses these tend to be filtered through two different processes: word frequency (removing very common and/or very rare words), and by filtering stop words (e.g. those included in the Natural Language Toolkit stop word list). This filtering step has been shown in multiple studies to influence the accuracy of document classification. Further, you do not do any form of lemmatization. Without this you are treating all surface forms of your words as unrelated, thereby increasing the sparsity of your tf-idf matrix and increasing the number of features for your SVM. I suggest you explore the impact of different pre-processing steps as they could have a large impact on your model perfomance (see Cornford et al. 2022 for an example of using different combinations of pre-processing steps).


- Line 227: Linear SVMs tend to be robust in situations where you have a larger number of features (predictors) compared to the number of data points. However, this is likely to break down as you increase the ratio of features to observations. As you are using a tf-idf matrix to generate features, you will have a number of features equivalent to the number of unique terms (after pre-processing), plus the numbers of bigrams. I assume this is a very large matrix, but you do not currently provide information about the number of features in your model. This relates to my previous two points in that I suspect you have a very large number of features compared to the number of observations in your data. This could be one major driver of both your moderate predictive performance, and the potential over-fitting you allude to (Line 470). You may also want to try different classifiers beyond SVMs as this could have a large impact on your predictive performance.


- Lines 409-441: This is a critical result of your study and indicates that you likely have some strong bias in your training data that may be causing overfitting and poor performance. When fitting your models you say you account for imbalance in your data labels (good!), but it seems you also have some imbalance in the taxa and ecosystems you have in your training data. As the point of your study is to find any “relevant datasets” which are agnostic to the taxa or ecosystems studied, you need to adjust your training data to account for this. One way is to gather more labelled training data (again goes back to limiting your searches to 50). Another way is to filter out taxonomy and ecosystem related terms from your feature set. This could be as easy as thinking of a manual list of stopwords and remove these as another pre-processing step. A more objective approach would be to identify an external list of taxon + ecosystem related terms (perhaps from an ontology or species lists, or list of defined eco regions/systems) and use these as stopwords. By filtering these out of your vocabulary before calculating the bigrams and tf-idf, the models will longer use them for training. As these seem to be relatively important for the SVM, removing them should improve your recall. This will ultimately force the model to focus on the more meaningful terms you highlight, while also hopefully limiting the model from picking up the underlying bias between taxa/ecosystem and relevance.


- Line 325-326: I am still uncertain about how the Quebec case study fits into this analysis. This line references you matching “Quebec” with affiliations of the authors, but this is not described in your methods. Further, I cannot see host the ATLAS project data compares with those you identified in your study and there is little discussion of the Quebec case study in the discussion. Why were the 616 ATLAS source datasets not used for training or at least some sort of comparison? As it stands, other than forcing of Quebec into the search terms (did you do this?), it is unclear how this is a case study for biodiversity monitoring in Quebec specifically.


One final thought is that you have these EBVs coded for each dataset, but you are only training a model on your composite relevance score. It may be useful for your framework to fit separate models based on the relevant quality of individual EBVs of interest. Again, your sample size may be too small for this, but you may find that since particular EBVs are associated with particular vocabulary, these text-based classification models may be better able to identify them. But if your relevant articles are relevant for different reasons (e.g. different EBVs), then you may not expect the same vocabulary to be important, leading to poor performance on the composite measure as compared to individual EBVs. Just a thought, but this approach could lead to more useful predictions for identifying future datasets (e.g. being able to explicitly identify spatial scales, or multi-species datasets, etc.)




Minor comments:

- First paragraph of the introduction can be further streamlined.

- Line 123-125: You say your approach provides “key insights into the effectiveness of search algorithms”, and you highlight the strengths and weaknesses of each, but it isn’t clear that you actually did this. You took the search results directly from Semantic Scholar and heavily subsampled them to conduct your study. Without looking further into the other search results returned and explicitly analyzing them, I don’t think this claim is valid.

- Throughout your introduction and methods you say you used repositories and APIs (in pural), but you only use the Zenodo API.

Line 155: Why is this algorithm “automated”? You actively train and test a classification model, so this process is not explicitly automated. Perhaps once a reliable and accurate model is trained, you may deploy it to eventually build an automated system to identify and extract data, but this is not the focus of your study.

- Line 167: How were these keywords chosen? In Fig. S1 it looks like you added “Quebec” to the keywords, but this is not reflected in Fig. 1.

- Line 189: You use the acronym EBV here, but only define it later (Line 207), and re-define it on line 297.

- Line 309: Again, I do not find the comparison between Semantic Scholar and the repositories to be a fair one if you heavily restrict the number of search outputs you investigate.

---

## Round 0.3 · accepted · Accept

Thank you for implementing those final amendments to the manuscript in response to the reviewer. I couldn't get his/her comments again this time, but from my own reading I believe authors have successfully addressed all issues pointed out previously.